# Effect of Ewe Diet on Milk and Muscle Fatty Acid Composition of Suckling Lambs of the Protected Geographical Origin Abbacchio Romano

**DOI:** 10.3390/ani10010025

**Published:** 2019-12-20

**Authors:** Isa Fusaro, Melania Giammarco, Matteo Chincarini, Michael Odintsov Vaintrub, Alberto Palmonari, Ludovica Maria Eugenia Mammi, Andrea Formigoni, Lorella Di Giuseppe, Giorgio Vignola

**Affiliations:** 1Faculty of Veterinary Medicine, University of Teramo, Località Piano D’Accio, 64100 Teramo, Italy; ifusaro@unite.it (I.F.); mchincarini@unite.it (M.C.); modintsovvaintrub@unite.it (M.O.V.); gvignola@unite.it (G.V.); 2Department of Veterinary Medical Science, Alma Mater Studiorum University of Bologna, via Tolara di Sopra 50, 40064 Ozzano Emilia, Bologna, Italy; alberto.palmonari2@unibo.it (A.P.); andrea.formigoni@unibo.it (A.F.); 3Faculty of BioScience and Technology for Food, Agriculture and Environment, University of Teramo, Via Renato Balzarini 1, 64100 Teramo, Italy; ldigiuseppe@unite.it

**Keywords:** lipid profile, sheep feeding, Abbacchio Romano, lamb performance, linseed-enriched diet

## Abstract

**Simple Summary:**

Consumers are increasingly aware of the nutritional quality of lamb products, especially in developed countries. Healthier lipid profiles might increase lamb meat consumption by concerned consumers. Pasture diets provide a viable option to enrich milk and meat products with fatty acids beneficial for human health. However, in Mediterranean areas, pasture is not available throughout the year, which means that weaned lambs are fed on concentrates. This investigation aimed to implement feeding strategies in suckling lamb to enhance healthier fatty acids in milk of dams and consequently in lamb’s meat by applying extruded linseed in a total mixed ration or using pasture. The proposed feeding plans were suitable to increase the n-3 fatty acids (FA) profile in milk and thus the lamb’s meat sourced from fresh pasture and linseed-enriched diets. Indoor rearing could include feeding lambs with linseed to help maintain a high level of beneficial fatty acids in lamb meat better than an un-supplemented diet or when pasture is not available.

**Abstract:**

Consumers increasingly pay more attention to the lipid profile of meat products and consume less meat to reduce cholesterol and heart disease. In Italy, sheep producers are increasingly feeding sheep fresh forage. We investigated whether the supplementation of dam diet with extruded linseed would be an alternative strategy to pasture for improving the intramuscular and subcutaneous FA compositions of their suckling lambs. The ewe diets were enriched with either extruded linseed (L), un-supplemented farm diet (F), or pasture (P). Milk saturated fatty acids (SFA) decreased in P and L compared with F, while the opposite pattern was observed for polyunsaturated FA (PUFA) and conjugated linoleic acids after seven days. The FA composition of lamb meat was similar to that of their dam’s milk, showing higher PUFA in P and L compared to F, while SFA was higher in F. Regarding the lamb meat obtained from barn-held ewes, L had lower n-6/n-3 content compared to F, while an intermediate content was found in P. These results indicate a better n-3 FA profile in milk and lamb’s meat from pasture and linseed-enriched diets. No changes in lamb performance were observed.

## 1. Introduction

In recent years, consumers have stimulated research for producing quality products guaranteed by rigorous certified production, certificates, and methods that respect the environment. To protect its gastronomic and cultural heritage in a global market, the European Union has established two protection systems known as protected designation of origin (PDO) and protected geographical indication (PGI). These systems were first regulated by the EC Regulation no. 2081/92 and then by Regulation EC no. 510/2006. In Italy, there are two lamb products registered under PGI: Sarda Suckling Lamb and Roman Lamb (Abbacchio Romano). The Roman Lamb represents 11–15% of the national meat market; they are slaughtered very young at about 10–12 kg after a suckling period of 20–30 days in order to minimize the milk loss for cheese production and also because Italian consumers prefer this kind of product. From birth to slaughter, lambs are managed with their mothers and are fed almost exclusively on milk. Suckling lambs are functionally non-ruminants, and their meat FA profile should reflect the FA profile of the suckled milk [1,2]. Therefore, changes in milk FA composition due to supplements in the dam diet can induce important differences in the FA profile of the meat and fat depots of the suckling lamb [3,4]. Several strategies have been tested in recent years to improve the fatty acids (FA) profile of ewe fat, like the use of fresh pasture or linseed supplementation. Fresh pasture has been shown to be an excellent source of polyunsaturated fatty acids (PUFA) to increase these FA in milk [5] and, subsequently, in suckling lamb meat fat [6]. Unfortunately, in Italy, like in other Mediterranean countries, fresh pasture is not available all year, and in this case, linseed supplementation (oil or seed) is a reliable alternative feeding strategy to enrich the milk fat from ewes [5,7,8] with PUFA, Conjugated Linoleic Acid (CLA), or omega three (n-3), and for their suckling lambs.

Currently, consumers are aware that consuming lamb meat with high levels of saturated fatty acids (SFA) is negative from a nutritional perspective as advised by the Academy of Nutrition and Dietetics, which, from a dietetics perspective, promotes the reduction of SFA and favours higher polyunsaturated FA (PUFA) content [9]. This is based on clinical studies that have referenced the relationship between the presence of some dietary acids, such as medium-chain saturated fats (C12: 0, C14: 0, and C16: 0), serum cholesterol, and heart disease [10]. In general, the FA composition of lamb includes as much as 53% SFA, 34% monounsaturated FA (MUFA), and only 13% PUFA [10]. Livestock production shows a growing interest in improving the nutritional characteristics of the products, for example, by natural enrichment of the animal diet, which leads to food characteristics that are beneficial to human health [11]. This natural enrichment could be achieved through pasture management, including “grass finished” animals, or the addition of supplements, such as flaxseed and oils in animal diets. In recent years, several studies have investigated the effect of ewe diet on the meat quality of their suckling lambs [12], showing how the intramuscular FA composition of suckling lambs is partly related to the sheep feeding system [13] during gestation or lactation [14,15,16] or both. To the best of our knowledge, there are no studies that have compared the suckling lamb meat obtained from grazing dams and that from animals fed indoors with a supplemented or un-supplemented diet. 

The aim of this trial was to investigate whether the supplementation of dam’s diet with extruded linseed would be an alternative to pasture strategy for improving the intramuscular and subcutaneous FA compositions of their suckling lambs, without detrimentally affecting animal performance.

## 2. Materials and Methods

### 2.1. Animals, Experimental Diets, and Feeding Routine

The lambs were managed during the experiment according to the European Directive 2010/63/EU on the protection of animals used for scientific purposes [17] complying with the Italian Legislative Decree 26/2014, and were slaughtered according to the European Union Regulations (Council Directive 93/119 EEC) [18] on the protection of animals at the time of slaughter or killing.

The study was conducted at a farm located in the Latium region, province of Viterbo (Italy), over a period of 30 days. Two weeks before the expected date of parturition, 54 Comisana ewes were randomly divided into three homogeneous groups based on age (32 ± 2 months), body weight (47.5 ± 1.2 kg), and number of lactations (2.3 ± 0.5). The supply of experimental diets began at parturition. The selected ewes were randomly allotted to the following three experimental treatment groups (18 sheep per group):(1)P—pasture group: ewes had daily access to pasture for 22 h/day without supplementation (average stocking rate: 15 ewes/ha). The pasture primarily consisted of Sulla (*Hedysarum coronarium*) as well as oats (*Avena sativa*) and clover (*Trifolium incarnatum*) seeded the previous fall. This composition was typical for pastures in the Latium region.(2)F—farm group (un-supplemented TMR): ewes had no access to pasture but were housed in straw-bedded pens and received a winter farm ration, which was the same as that normally practised in Central Italy. The ingredients of the total mix ration (TMR) were grass hay at 1.100 and 0.800 kg/day of a concentrate-based meal (oat, barley, and soybean)(3)L—linseed-enriched group: ewes had no access to pasture but were housed in straw-bedded pens and received the same winter ration as group F. The ingredients of the TMR were grass hay at 1100 and 800 g/day of a concentrate-based meal with 0.190 kg of extruded linseed added. Linseed, ground to pass through a 4-mm screen, was extruded in a single screw extruder with a throughput of 1600 kg/h (barrel length: 3.2 m; die diameter: 7 mm; screw speed: 300 rpm; temperature at the end of the barrel: 130–138 °C; duration: 1 min). After extrusion, the product was dried in a counter flow cooler for 12 min.

The ingredients and the chemical compositions of the three diets are shown in Table 1. The groups also included 54 new-born lambs, one for each ewe, which were housed with their respective mothers in each group. The lambs were nourished by suckling from birth until 28 days of age, at which time, they were sent to an EU authorized slaughter facility. 

### 2.2. Milk Sampling and Composition 

Individual milk samples were collected from each ewe by parlour-milking at the beginning of the experimental period starting on day 4 (T0), 7, 14, and 28 after lambing.

On each milk sampling day, the ewes were separated from their lambs two hours before milking both in the morning and in the afternoon. The milk collected was sampled and refrigerated until the analysis. The chemical composition of each milk sample (fat, protein, lactose, and total solids) was determined using a Milkoscan 6000 instrument (Foss Electric, Hillerød, Denmark). Aliquots from each sample were stored at −80 °C for FA analysis. The milk lipid fraction was extracted according to the AOAC official method [19] for FA composition. FAME was separated using a gas chromatograph (Thermo Scientific, Waltham, MA, USA) equipped with a capillary column (Restek Rt-2560 column fused silica 100 m × 0.25 mm highly polar phase; Restek Corporation, Bellefonte, PA, USA) and a flame ionization detector. Hydrogen was used as the carrier gas. The initial holding temperature was 55 °C for 1 min, which was increased to 170 °C at a rate of 10 °C/min and held for 30 min. The final temperature of 215 °C was reached at a rate of 2 °C/min and was held for 4 min. Peak areas were quantified using ChromeCard software, and the relative value of FA was expressed as a percentage of the total FA. 

### 2.3. Slaughter Procedure and Carcass and Meat Measurements

Each lamb was weighed every week. The individual data thus obtained were used to calculate the average daily gain. The lambs were slaughtered at 28 days (±1.5 d) as a result of market demand owing to the Easter holidays. At the slaughterhouse, the live weight was recorded. The lambs were electrically stunned and slaughtered using standard commercial procedures for this species. The carcasses were weighted (hot carcass weight, HCW) according to the specification for the “Roman Lamb” (including head and pluck, without distal limb, dissected at the tarsometatarsal/carpal-metacarpal joints). After 45 min, pH (pH1) was measured in situ on the right side of each carcass between the fifth and sixth lumbar vertebrae [20] using a penetrating electrode adapted to a portable pH meter (Crison pHmetro 507 and a 52–32 spear electrode, Crison Instruments, Spain). The carcasses were stored in cold rooms at 4 °C, and after 24 h, carcasses were weighed (cold carcass weight, CCW), and pH was measured for the second time (pH2) as for pH1. The chilling losses were calculated as the difference between HCW and CCW.

Fat and meat colour parameters (L*, Chroma*, Hue°) were successively measured according to the CIELab system, using a Minolta Chroma Meter CR-300 (Minolta Camera Co., Osaka, Japan) with a D65 illuminant and an 8-mm aperture. Colour was measured for freshly cut surfaces, and they are reported as the mean of three measurements at the cross-section of the muscle. Conformation score (P− = 1 to E+ = 15) and fatness score (1− = 1 to 5+ = 15) were estimated according to the EEC guidelines. The firmness and colour of subcutaneous fat were judged on cooled carcasses according to the 4-point scale of the Institut de l’Elevage (1: very firm and white to 4: very soft, oily, and coloured; [21]). To determine the drip and cooking losses, meat samples of about 50 g each and roughly cubic in shape were taken from the longissimus lumborum muscle, as suggested by Honikel [22]. Further individual samples of loin muscle were analyzed to measure moisture, fat, protein, and ash [19].

### 2.4. Fatty Acid Composition of Meat 

From each carcass, samples of the longissimus lumborum were taken between the 6th and the 13th ribs, vacuum-packed, and stored at −80 °C until analysis. Before analysis, samples were thawed in tap water at 15–17 °C for 2 h. Intramuscular lipids were extracted following the protocol suggested by Folch et al., [23]. Following cold methylation of FAs with the technique proposed by Fraga and Lerker [24], the FA profile was determined from extracts by gas chromatography using a capillary column Chrompack CP-SIL 88 (100 m length, 0.2 μm film thickness, 0.25 mm inner diameter), working from 160 °C (1 min) to 175 °C (4 °C/min) for a total of 28 min, and then from 175 to 215 °C (5 °C/min) for 35 min with the injector at 250 °C and detector at 260 °C and using an ionized flame detector. Prior to statistical analysis, the data on FA composition were processed to calculate the following FA classes and indices: MUFA (FA with single double bond), PUFA (FA with more than one double bond); SFA (FA without double bonds); UFA (FA with one or more double bonds); n-3 (S C18:2 t11, c15 + C18:2 c9, c15 + C18:3 c9, c12,c15 + C:22:5 c7, c10, c13, c16, and c19 + eicosapentaenoic acid (EPA) + docosahexaenoic acid (DHA)); n-6 (S C18:2t9, t12 + C18:2 c9, t12 + C18:2 t9,c12 + C18:2 c9, c12 + conjugated linoleic acids (CLA) t10, c12 + C20:2 c11, c14 + C20:3 c8, c11, c14 + C20:4 c5, c8, c11, and c14). 

The atherogenic index was calculated according to Ulbricht and Southgate [25] as follows: (C12:0 + 4 × C14:0 + C16:0)/(MUFA + PUFA). The I-Harris index was calculated as the sum of EPA and DHA [26], and undesired trans FA (UTFA) as the sum of C18:1t9 + C18:2 t9t12 [27].

### 2.5. Statistical Analysis

Milk chemical composition was analyzed by GLM procedure considering dietary treatment as the fixed effect. Data on FA composition of milk were analyzed by mixed model repeated measures using the SPSS 13.0 statistical package (SPSS, 2006) [28], including in the model the fixed effects of dietary treatments, sampling time, and the interactions between them. The data collected on carcasses and meat were processed by GLM in SPSS, considering the fixed effect of dietary treatment. Tukey’s test was used to assess significant differences between means. 

## 3. Results 

### 3.1. Milk

The effects of the different diets on the chemical composition of milk are presented in Table 2. The experimental diets did influence milk composition: the protein content of the milk was higher in P compared to L and F (*p* < 0.001). In addition, dietary treatment determined significant differences (*p* < 0.05) among the milk total fat percentage from each group showing higher value for group L than F and lower fat percentage in milk from group P.

The FA profile of milk fat was modified by dietary treatment. Figure 1 shows the temporal changes in SFA, PUFA, CLA, and n-6/n-3 ratio when ewes were fed the three experimental diets. The proportion of SFA from 0 to 28 days of lactation increased significantly in group F compared to L and P (Figure 1), and differences among the groups were evident from milk samples after 7 days of lactation. In group F, the concentration of SFA increased after 7 days of lactation (*p* < 0.05) and increased further at 14 and 28 days of lactation (*p* < 0.01). The concentration of SFA is higher in all time points except for T0 in group F compared to L and P (*p* < 0.01).

The concentration of PUFA from 0 to 28 days of lactation increased significantly in groups L and P. Differences among the groups were evident from milk samples after 7 days of lactation. Figure 2 shows that the concentration of PUFA was similar between groups P and L in the second milk sample (7 days of lactation), but at 14 and 28 days of lactation, a higher PUFA concentration was observed in the milk of group L compared with that of group P (*p* < 0.05), while that of group F was lower than in both other groups at 7, 14, and 28 days of lactation. The proportion of CLA (Figure 3) significantly increased from 0 to 28 days of lactation in groups P and L compared with that of group F. Specifically, the proportion of CLA in the milk of group P shifted from an average of 0.67% at 7 days of lactation to 0.83% at 28 days of lactation. The concentration of CLA in group L was higher than that in group F at 7, 14, and 28 days of lactation but was lower than that in group P (*p* < 0.01). At 7, 14, and 28 days of lactation, the ratio of n-6/n-3 was higher in group F compared with that in group L, while that of group P was intermediate between the other two groups (*p* < 0.01). Specifically, the lower ratio was observed in group L with 1.86% and 1.18% at 14 and 28 days, respectively, compared with group P, which showed n-3/n-6 ratios at the same time points of 2.96% and 2.39%, respectively (*p* < 0.05).

### 3.2. Lamb Performance and Carcass Characteristics

The average daily gain (Table 3) showed that the lamb growth performance was similar among the groups. The average CCW was 6.74, 6.90, and 6.77 kg, respectively, for groups F, L, and P, showing no differences with dietary treatment. No differences were observed regarding the conformation score, fatness score, or fat softness score (*p* > 0.05), while the fat colour score was higher in group P than in groups L and F (*p* < 0.05).

### 3.3. Meat Quality of Suckling Lambs

Meat and fat characteristics in response to the three dietary treatments are shown in Table 4. Meat and fat lightness L* were similar in the three groups, while a* was comparatively higher in group P than in groups F and L (*p* < 0.001) for meat and fat (*p* < 0.05). Another factor that was not influenced by dietary treatment was pH at both 45 min and 48 h after slaughtering (*p* > 0.05). The drip and cooking losses were also not significantly different among the experimental groups. 

### 3.4. Fatty Acid Composition of Intramuscular Fat in Relation to Different Dam Dietary Treatments

Table 5 shows the effects of dam dietary treatment on the FA composition of the suckling lamb longissimus lumborum. The results showed that groups L and P were different from group F, especially for the most important FA classes (Table 5). Specifically, the lambs in groups L and P exhibited higher percentages of PUFA (*p* < 0.05), CLA (*p* < 0.05), EPA and DHA (I-HARRIS index) (*p* < 0.05) than the animals in group F. Conversely, the concentration of SFA (*p* < 0.05) and n-6/n-3 ratio (*p* < 0.05) in muscle was higher in lambs of group F than in those of groups P and L. About the index of desaturation, the diets did not seem to significantly influence the variability in C16:1/C16:0 and C18:1cis/C18:0 ratios in muscle. Furthermore, the myristoleic/myristic acid ratio showed no differences among the groups.

## 4. Discussion 

### 4.1. Milk Composition 

The composition of milk was different among the three experimental groups. Luna et al. [29] reported a decrease in milk fat and no difference in milk yield in ewes fed a supplement enriched with whole linseeds compared to a control diet by Gómez-Cortés et al. [30], in an experiment on sheep milk in which the ewes were fed extruded linseed (6% or 12% of the diet), who reported a significant effect of supplementation and overall production of milk on the percentage of lipids, proteins, and lactose. This effect was attributed to increased energy availability owing to the lipids used in the rations. In our study, the experimental time probably influenced the productive performance of the sheep. In fact, in Gómez-Cortés [30], the integration with extruded linseeds lasted for 60 days, a period longer than our experimental time. In our work, we did not observe any effect on milk yield but higher average lipid levels in the group reared inside and fed a farm diet or a diet with linseeds added compared with that of the pasture grazed by the ewes. However, our results are also in agreement with those of Gallardo et al. [16], who suggested that the response of sheep to supplementation with high concentrations of lipids, rich in PUFAs, and the generation of these isomers (involved in milk fat depression) did not significantly change milk fat in ewes.

In bovine species, several reports have highlighted a significant reduction in the lipid component of milk as a result of the integration of a non-protected lipid diet [31]. However, findings in sheep [16,32] seem to indicate a lack of this negative effect, which is associated with the use of fat in ewe diets. In our study, a higher fat concentration in the milk of groups L and F could also be influenced by the higher content of NDF in the diet of ewes reared inside compared with grazing ewes. Indeed, a high NDF content plays a fundamental role in the acetate:propionate ratio and, consequently, in the synthesis of milk lipids. For example, Cannas [33] recommended that the levels of NDF in the diet of dairy sheep should not be lower than 33 g of NDF/100 g of DM, with an optimal range of NDF from 45 to 33 g/100 g of DM. Our results completely agree with this recommendation. Supplementation with extruded linseed increased milk fat and decreased milk protein as often happens with protein sources. The lower concentration of lignin and the sugar content of the pasture might have stimulated microbial activity and thus resulted in higher milk protein compared to F and L.

### 4.2. Milk Fatty Acid Composition

Figure 1 compares the main classes of FA in the milk of ewes treated with the three different diets at different sampling times. The concentration of SFA differed during the 28 days of feeding the experimental diets. As Figure 1 shows, the concentration of SFA increased gradually from 60.51% to 71.69% at 28 days of lactation (*p* < 0.05) in group F. A diet without an appropriate supplementation with PUFA usually increases SFA from C8:0 to C16:0 in the milk of ewes [29] and goats [34]. As was logical to expect, in the P and L groups, the concentration of SFA decreased between the times.

Throughout the 28-day experiment, the concentration of PUFA in the milk of dairy ewes was altered by diet. Most previous studies on dairy ewes have evaluated the PUFA production in grazing animals or animals fed diets based on hay and concentrates or enriched/un-enriched with seeds, such as sunflower or linseed, or by-products. To the best of our knowledge, the present work is the first to compare the quality of products from animals fed a linseed-supplemented diet, farm diet without integration, and pasture-based diet. Similar to other authors, we observed a shift in PUFA production 7 days after the introduction of extruded linseeds as a dietary supplement. Even the in vitro study by Jouany et al. [35] demonstrated that the changes in the accumulation of biohydrogenation intermediates after oil inclusion in media occur after only 30 min. The ruminal microbes are probably able to adapt rapidly to the biohydrogenation of unsaturated FA because they are in contact with them in most ruminant diets. 

In the present study, the temporal patterns of PUFA in milk were similar to those reported by Luna et al. [29], who observed an increase in the principal PUFA, like C18:2 c9 and C18:3 n-3, in the milk of ewes supplemented with whole linseeds (17% DM) in the early days post supplementation and then a decrease after the 10th day. In our study, we observed the same trend: after 7 days of integration, the concentration of PUFA was significantly higher in groups P and L compared to group F (*p* < 0.05). At 14 days after birth, the PUFA concentration was higher in group L compared with groups P (*p* < 0.01) and F (*p* < 0.01) and remained higher until the last milk sample at 28 days. The results of our trial are in accordance with those of previous studies that reported higher proportions of PUFA in milk from grazing ewes than that from indoor-fed ewes without supplements [6]. Regarding the higher concentration of PUFA in group L compared with that in group P at 14 and 28 days of lactation, our hypothesis states that the long-chain FA precursors could have greater accessibility to ruminal bacterial enzymes when diets are supplemented with linseeds [12] compared with feeding the animals by pasture.

Figure 3 shows that a higher concentration of CLA at the final time point (28 days of lactation) was observed in the milk of group P (0.83%) compared with that of group F (0.44%) (*p* < 0.05), while group L had a lower CLA concentration (0.66%) than group P but higher than group F (*p* < 0.05). 

The effect of linseed meal in sheep milk was reported in other publications as well [6,30]. In our trial, ewes were fed experimental diets for 28 days, whereas the other studies used longer feeding periods, for example, Gómez-Cortés et al. [30] fed ewes for 60 days, while Mele et al. [8] fed them for 10 weeks. We think that the higher level of CLA in group P than group L at 14 and 28 days of lactation was probably due to the linoleic acid in the pasture diet being easily accessible to ruminal microbiota or restricting the activity of *Butyrivibrio fibrisolvens* differently from the other two diets. *B. fibrisolvens* is responsible for the hydrogenation of linoleic acid and linolenic to vaccenic acid [36]. 

Figure 4 illustrates the ratio between n-6/n-3. This ratio evidenced the higher concentration of PUFA omega-3 in groups P and L compared with that of group F. In particular, the high levels of dietary lipids contained in linseeds induce ruminal adaptations that lead to alterations in the formation of specific intermediates that influenced the final concentration of omega-3 in the milk more than the pasture diet. Dairy ewes were grazing pasture in the vegetative stage and rich in PUFA, and this probably enhanced the PUFA in their milk [37]; however, this study demonstrated that, in 28 days, linseed supplementation was more efficient in producing milk enriched with PUFA n-3 compared with the pasture diet.

### 4.3. Suckling Lamb Performance

The lambs in the present work were fed with maternal milk and exhibited no differences in their performances (Table 3), even though the concentration of protein and fat in the milk of dams were different between the groups (Table 2). Previous research has shown that changes in suckling lamb performance are mainly related to differences in milk yield, milk fat, and protein levels. The lambs in the present work were fed maternal milk and exhibited no differences in their performances (Table 3) even though the concentration of protein and fat in the milk of dams were different between the groups (Table 2). Previous research has shown that changes in suckling lamb performance are mainly related to differences in milk fat and protein levels [4]. In our work, the time of suckling (28 days) was probably too short to have an effect on lamb performances owing to the different concentrations of fat and protein in the milk. Moreover, since all the ewes had only singles (the number of male and female lambs was balanced between treatments), the milk produced by the mother could have been different among treatments with lowest milk production. If milk is not all suckled, the mothers quickly adjust their production to the amount used by their lambs, so the amount measured could have been affected by the low demand for milk. Otherwise, no detrimental effect of linseed supplementation on growth performance has been reported by most studies on lambs, regardless of the form of seed or oil [38,39] In the present study, the carcass conformation score was not different among the three experimental groups. Previous studies have reported inconsistent results regarding the effect of diet on conformation and fatness scores. Demirel et al. [39] reported lower conformation scores and higher fatness scores in lambs fed extruded linseed during the post-weaning period. However, no such effect on conformation or fatness was reported by Radunz et al. [40] using 3% oil in the diet (soybean and linseed) in pre- and post-weaning lambs. Furthermore, Guerrero et al. [41] reported that carcass traits were unaffected by 5–10% and 15% linseed dietary supplementation. Instead, Borys et al., [42] found a decreased content of fat in the muscles of suckling lambs that received milk enriched with CLA, which slows down the process of fat synthesis in animal bodies. Otherwise, our results showed that milk quality did not influence lamb growth performance or fat deposition at 28 days, probably due to short supplied time. 

The most significant difference observed in meat characteristics was in the instrumental colour, for which, only the a* was comparatively higher in the lambs of group P compared to those of groups F and L for meat (*p* < 0.01) and fat (*p* < 0.05). Similarly, all diets resulted in meat that conformed to the norm for colour, i.e., luminosity of 40 for all groups. This value is considered to be the threshold of meat acceptability by consumers [43]. Meat and fat colour, besides live weight, depends on age, exercise, and nutrition. In the present study, the lambs ate only dam milk and were slaughtered at the same age and weight; thus, we considered that the colour of the meat was influenced by the dam diet. In fact, nutritional background is one of the main factors influencing meat characteristics like colour and the effect of pasture is evident, and it is clearly demonstrated in the literature that meat from animals finished on pasture is redder than meat from animals finished on concentrate [44]. 

### 4.4. Intramuscular Fatty Acid Composition 

The meat FA profile was similar to the FA profile of the milk of their dams. This relationship has previously been described in suckling lambs [4,6,42]. In lambs at 28 days of age, the rumen is not yet functional, so the milk FA are absorbed directly by the intestine without ruminal biohydrogenation activity. It was not surprising that changes in dam milk FA composition due to diet can induce significant differences in the FA profile of meat and fat deposits in their suckling lambs. In the young lambs, as with other suckling animals, the essential FAs were incorporated directly into the muscle rather than being stored in the adipose tissue, which is considered to be an important metabolic role. This is because the intramuscular fat of pre-ruminal lambs is more abundant in PUFA compared with that of adults [4] and is based on a higher phospholipids proportion in these deposits [45]. As showed by Nudda et al., [46], there is a strong correlation between the FA in milk and its concentration in the meat of suckling animals. Furthermore, Nudda et al. found a high correlation between meat CLA of suckling goat and their mother’s milk.

The meat from ruminants has often been considered to be detrimental to human health owing to its high SFA content [47]. However, it is possible to reduce the concentration of SFA in ruminant meat products by enhancing the polyunsaturated fatty acids concentration of PUFA in their rations.

In fact, as shown in Table 5, the lower SFA content in groups L and P guarantees healthier meat products compared with that in group F. We should consider from a human health perspective that adding a source of PUFA could be sufficient to reduce SFA in dam diets and improve meat quality. The concentration of SFA in the L group is consistent with the results of a previous study based on observations of suckling lambs of the same weight [13].

Linseed supplementation in ewe diets increased the PUFA concentration in the meat of suckling lambs in group L (21.91 mg/100 g FA) more than in those in groups P (19.90 mg/100 g FA) and F (17.75 mg/100 g FA) (*p* < 0.05). These results agree with those of Cabiddu et al. [37]. Dairy products enriched with n-3 FA are considered to represent healthy foods by the scientific literature [47]. Studies on humans have indicated an association between dietary intake of dairy products naturally enriched with PUFA and the prevention of cardiovascular disease CVD through decreased levels of plasma cholesterol [47]. Among long-chain PUFA, n-3, EPA (20:5 n-3), and DHA (22:6 n-3) have been recognized for their beneficial effects on the cardiovascular system and on the brain and visual system during fetal development and throughout life [1]. However, the concentrations of EPA and DHA are highly variable in meat products compared with those in fish or vegetable oils [48]. Our results also showed a higher I-Harris index (sum of EPA and DHA) in groups P and L compared with group F. This is an important result if we consider that a meat-based diet containing lamb meat caused a major improvement in the health conditions of allergic infants [49,50]. The effectiveness of a home-made lamb meat-based formula in babies with food-induced atopic dermatitis was detailed in the review of [51]. Therefore, the enrichment of lamb meat with PUFA n-3 and CLA could be of great nutritional interest [52].

The higher concentration of CLA in the lamb meat of group P compared with that of group L (*p* < 0.05), could be due to the higher concentration of the principal precursor of CLA, C18:1 trans-11, in the pasture diet compared to TMR with and without supplementation. However, the concentration of CLA in the meat from group L was significantly higher than that observed in the meat from group F. the pasture diet was not a stable source of these FA, and meat FA quality tends to vary during the year following seasonal grazing quality and availability. A previous study confirmed that a pasture diet is a good source of CLA and PUFA when the pasture is young and in the vegetative phase [37]. These results demonstrate that the uses of linseeds or pasture are able to improve the concentration of CLA in suckling lamb meat compared to an un-supplemented diet. It must be considered that the CLA may also originate from endogenous synthesis by the enzymatic activity of stearoyl Co-A desaturase (SCD), which is able to insert a double bond in the cis-9 position. Since the activity of this enzyme in tissue samples is complex and expensive to investigate, Palmquist et al. [53] proposed the following ratio to determinate SCD activity: C14: 1 / C: 14: 0; C16: 1 / C: 16: 0; C18: 1cis9 / C: 18: 0. For meat tissue in particular, the ratio C14: 1 / C: 14: 0 seems to better indicate the activity of SCD. We did not observe a significant difference in this ratio in the lamb meat among the three experimental groups. We assumed that the increase in CLA in the meat of grazing lambs depended only to the dam’s milk. A period of 28 days was enough to guarantee the transfer of CLA from milk to the lamb meat. Furthermore, Scerra et al. [6] reported that the endogenous synthesis of CLA was higher in heavy lambs compared with that observed in lighter lambs.

Regarding the lamb meat obtained from barn-held ewes, group L exhibited a lower n-6/n-3 ratio highlighting a better n-3 FA profile in the milk, and consequentially, in the lamb meat, sourced from the fresh pasture and linseed-integrated diets. This result agrees with the findings of Scerra et al. [6], who reported that the n-6/n-3 ratio in lamb meat increased with the pasture-based diet compared with a TMR. Additionally, Gómez-Cortés et al. [30] found a lower n-6/n-3 ratio in the meat of animals fed linseeds in accordance with our results. According to Cabiddu et al. [37], animals fed diets based on herbage only, similar to the pasture-fed group of our study, had a significantly lower level of UTFA (0.29 mg/100 g of FA) compared with L (0.49 mg/100 g of FA) and F (0.73 mg/100 g of FA) (*p* < 0.01), probably because linseeds are rich in the precursors of these putatively noxious FAs. Several studies have highlighted the importance of not only increasing the content of health-promoting FAs but also decreasing the level of potentially harmful saturated and trans FAs, such as UTFAs (C18:1 t9 + C18:2 t9, t12) when developing human health care products [27,37,54]. A positive association between increased circulating low-density lipoprotein (LDL) levels and UTFA intake has been shown to alter or promote the relationship between the LDL atherosclerosis and coronary artery disease [54].

## 5. Conclusions

The results obtained in this study indicate that the FA composition of suckling lambs of the protected geographical origin Abbacchio Romano was influenced by the composition of maternal milk for 28 days of rearing. The most interesting aspect of the present study was that supplementation with extruded linseeds in the diet of dams was able to increase the levels of PUFA, particularly n-3 FA, in lamb meat in comparison with pasture-based and typical farm diets. However, the levels of CLA in lamb meat achieved using this approach were lower than those from a grazing diet. This study demonstrated that supplementing sheep diets with linseeds improved the FA profile of lamb meat and could be a beneficial feeding strategy to apply in ewe farms when fresh pastures are not available. Nevertheless, pasture is a viable option relative to an un-supplemented diet because it combines high product quality with low feeding costs. Our results seem to indicate that supplementation with extruded linseeds helps to maintain a high level of beneficial FAs in lamb meat and it is recommended when pasture is not available as occurs in some regions of Italy during the winter.

## Figures and Tables

**Figure 1 animals-10-00025-f001:**
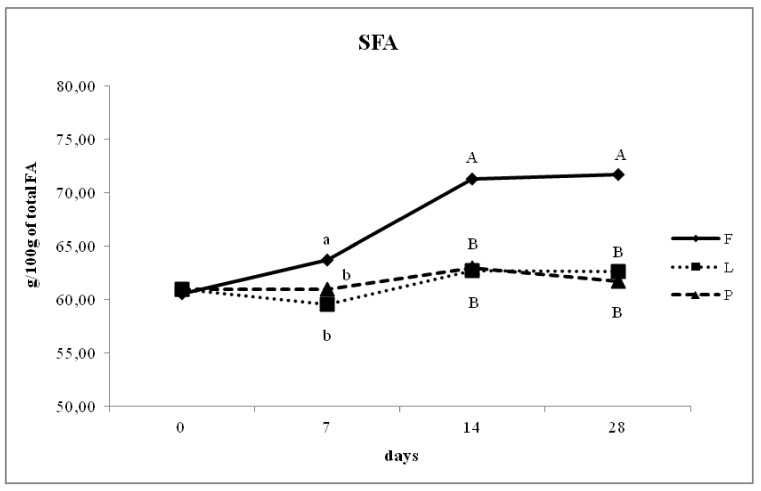
Day-dependent changes of saturated fatty acids (SFA) in the milk of ewes fed un-supplemented diet (F), diet supplemented with extruded linseed (L), and pasture-based diet (P). Different letters indicate significant differences (a, b, c: *p* < 0.05; A, B, C: *p* < 0.01).

**Figure 2 animals-10-00025-f002:**
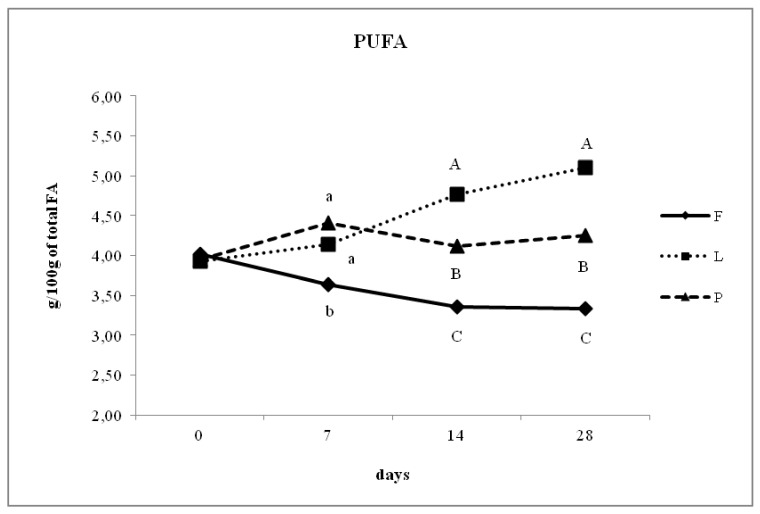
Day-dependent changes of polyunsaturated fatty acids (PUFA) in the milk of ewes fed un-supplemented diet (F), diet supplemented with extruded linseed (L) and pasture-based diet (P). Different letters indicate significant differences (a, b, c: *p* < 0.05; A, B, C: *p* < 0.01).

**Figure 3 animals-10-00025-f003:**
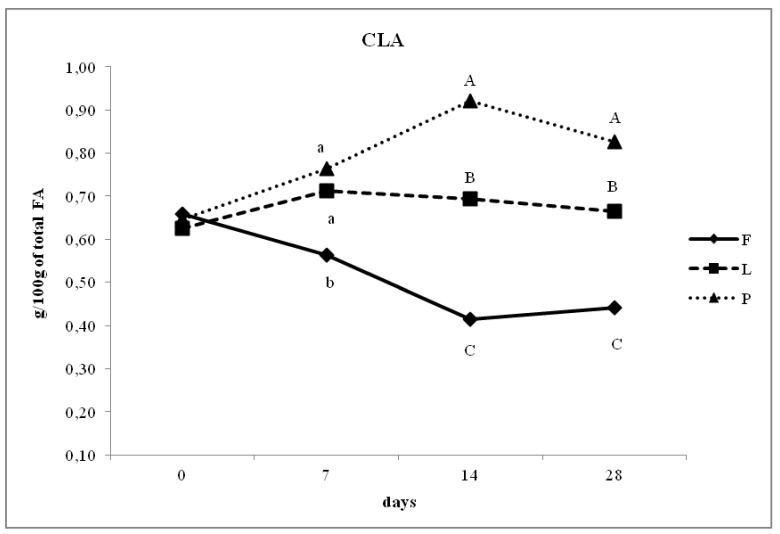
Day-dependent changes of Conjugated Linoleic Acid (CLA) in the milk of ewes fed un-supplemented diet (F), diet supplemented with extruded linseed (L) and pasture-based diet (P). Different letters indicate significant differences (a, b, c: *p* < 0.05; A, B, C: *p* < 0.01).

**Figure 4 animals-10-00025-f004:**
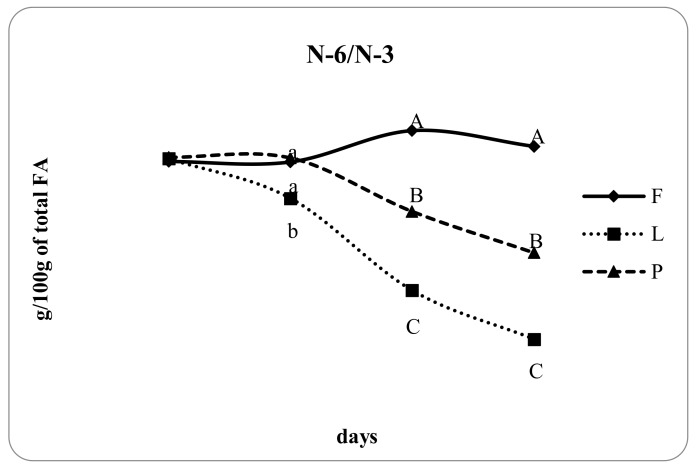
Day-dependent changes of n-6/n-3 in the milk of ewes fed un-supplemented diet (F), diet supplemented with extruded linseed (L), and pasture-based diet (P). Different letters indicate significant differences (a, b, c: *p* < 0.05; A, B, C: *p* < 0.01).

**Table 1 animals-10-00025-t001:** Ingredients (% DM), chemical composition and fatty acid (FA) composition of the experimental diets.

Ingredients	Dietary Treatment ^1^		
	F	L	P
Grass hay	56.79	56.79	….
Oatmeal	12.91	10.33	….
Barley meal	12.91	10.33	….
Soybean meal	15.85	11.20	….
Extruded linseed	….	9.81	….
Mineral and vitamin mix	1.55	1.55	….
*Chemical composition*			
DM	89.12	90.56	16.72
CP	20.71	19.97	19.57
EE	2.86	4.83	2.50
NDF	39.24	40.75	29.77
ADF	25.48	25.2	21.91
ADL	3.68	3.93	1.41
*Fatty acid profile (% of total FA)*			
C12:0	0.15	0.18	0.17
C14:0	0.6	0.2	0.30
C16:0	18.8	8.8	12.7
C16:1	1.01	0.98	1.1
C16:1c9	0.3	0.1	0.2
C18:0	2.6	4.1	1.11
C18:1c9	23.8	21.1	1.81
C18:2c9, c12	40.0	13.3	11.5
C18:3c9, c12, c15	11.7	50.6	70.3
C18:1c11	1.0	0.6	0.8

^1^ F: un-supplemented diet; L: diet supplemented with extruded linseed; P: pasture-based diet.

**Table 2 animals-10-00025-t002:** Milk chemical composition at the end of the experimental period.

Item	Dietary Treatment ^1^			*p* Value	SEM
	F	L	P		
Fat (%)	4.89 ^b^	5.21 ^c^	4.27 ^a^	<0.05	0.30
Protein (%)	4.76 ^B^	4.55 ^B^	5.37 ^A^	<0.001	0.10
Lactose (%)	4.58	4.52	5.01	0.435	0.17
Total solids (%)	14.47	14.69	15.54	0.558	0.44

^1^ F: un-supplemented diet; L: diet supplemented with extruded linseed; P: pasture-based diet. Different letters in the same row indicate significant differences (a, b, c: *p* < 0.05) (A, B, C: *p* < 0.001).

**Table 3 animals-10-00025-t003:** Effect of experimental ewe diets on growth performance and carcass traits of suckling lambs.

Item	Dietary Treatment ^1^			*p* Value	SEM
	F	L	P		
Birth weight (kg)	4.87	5.05	4.97	0.23	0.96
Pre-slaughtered weight (kg)	10.72	10.92	10.78	0.86	0.25
Age at slaughter (d)	28.2	27.5	28.7	0.26	1.01
Daily gain (g)	221	246	229	0.90	1.12
Cold carcass weight (kg)	6.74	6.90	6.77	0.84	0.98
Dressing (%)	51.94	53.07	50.28	0.60	0.45
Conformation score ^2^	8.5	8.9	8.4	0.21	1.23
Fatness score ^3^	7.3	6.9	7.1	0.32	0.45
Fat softness score ^4^	3.0	2.9	3.2	0.45	0.87
Fat colour score ^5^	1.5 ^a^	1.2 ^a^	2.8 ^b^	<0.05	0.36

^1^ F: un-supplemented diet; L: diet supplemented with extruded linseed; P: pasture-based diet. ^2^ 15 points conformation scale (P− = 1 to E+ = 15). ^3^ 15 points fatness scale (1− = 1 to 5+ = 15). ^4^ 4 points scale: 1: very firm to 4: very soft and oily fat. ^5^ 4 points scale: 1: white to 4: coloured fat. Different letters in the same row indicate significant differences (a, b, *p* < 0.05).

**Table 4 animals-10-00025-t004:** Effect of experimental ewe diets on pH, colour, cooking, drip losses, and chemical composition of suckling lamb meat.

Item		Dietary Treatment ^1^			*p* Value	SEM
		F	L	P		
pH_1_		6.67	6.75	6.78	0.32	0.03
pH_u_		5.56	5.49	5.54	0.42	0.02
Meat colour ^2^						
L*		52.43	53.27	49.16	0.14	0.85
a*		12.29 ^B^	10.70 ^B^	14.33 ^A^	<0.001	0.26
b*		8.08	6.87	6.39	0.21	0.38
Fat colour ^2^						
L*		68.5	67.7	69.1	0.23	0.96
a*		4.65 ^b^	5.54 ^b^	6.33 ^a^	<0.05	0.36
b*		7.12	7.56	8.16	0.45	0.48
Water holding capacity						
Drip loss	%	3.38	3.58	3.60	0.74	0.13
Cooking loss	%	19.79	16.91	20.55	0.37	1.07
Chemical composition						
Moisture	%	74.79	75.43	75.45	0.72	0.33
Protein	%	21.70	21.05	21.02	0.66	0.31
Fat	%	2.29	2.36	2.35	0.98	0.14
Ash	%	1.22	1.16	1.18	0.68	0.02

^1^ F: un-supplemented diet; L: diet supplemented with extruded linseed; P: pasture-based diet. ^2^ Colour values measure: L* = darkness to lightness, a* = degree of redness, b* = degree of yellowness. Different letters in the same row indicate significant differences (a,b: *p* < 0.05) (A,B,: *p* < 0.001).

**Table 5 animals-10-00025-t005:** Sum of fatty acids (expressed in g/100 g) and nutritional values of meat from lambs fed milk of ewes that received three different diets.

Fatty Acids	Dietary Treatment ^1^			*p* Value	SEM
	F	L	P		
Principal FA categories					
SFA	44.16 ^c^	40.95 ^a^	42.08 ^b^	<0.05	0.12
MUFA	37.09	37.14	38.01	0.77	0.36
PUFA	18.75 ^a^	21.91 ^c^	19.90 ^b^	<0.05	1.25
UFA	56.84 ^a^	59.05 ^b^	57.91 ^b^	<0.05	0.89
CLA	0.47 ^a^	0.69 ^b^	0.80 ^c^	<0.05	0.36
n-3	3.09 ^a^	4.80 ^c^	3.67 ^b^	<0.01	1.02
n-6	16.19	16.54	14.43	0.25	0.77
UTFA	0.73 ^c^	0.49 ^b^	0.29 ^a^	<0.05	0.37
Nutritional index and ratio					
SFA/UFA	0.77	0.69	0.72	0.06	0.45
n-6/n-3	5.30 ^c^	3.65 ^a^	4.30 ^b^	<0.05	0.69
TI	0.98	0.88	0.84	0.22	0.89
AI	0.76	0.66	0.86	0.24	1.04
I-HARRIS	1.56 ^a^	1.73 ^b^	1.49 ^c^	<0.05	0.81
DI C14:0	3.49	3.67	3.78	0.35	0.23
DI C16:0	6.71	5.55	6.34	0.23	0.87
DI C18:0	73.29	69.61	73.80	0.65	0.98

^1^ F: un-supplemented diet; L: diet supplemented with extruded linseed; P: pasture-based diet. Different letters in the same row indicate significant differences (a, b, c: *p* < 0.05). The data correspond to the analysis of fresh meat from L Lumborum of Roman Lamb SE: standard error; SFA = saturated fatty acid; MUFA = monounsaturated fatty acid; PUFA = polyunsaturated fatty acid; UFA: unsaturated fatty acid; n-3=Σ C18:2 t11, c15 + C18:2 c9, c15 + C18:3 c9, c12,c15 + C:22:5 c7, c10, c13, c16, c19 + EPA + DHA; n-6 = (S C18:2t9, t12 + C18:2 c9, t12 + C18:2 t9, c12 + C18:2 c9, c12 + CLA t10, c12 + C20:2 c11, c14 + C20:3 c8, c11, c14 + C20:4 c5, c8, c11, c14; AI: Atherogenic Index; Thrombotic index: TI; CLA = conjugated linoleic acid; I-Harris = (EPA+ DHA); UTFA = (C18:1 t9 + C18:2 t9, t12).

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
