# Peer review of "Effect of Ewe Diet on Milk and Muscle Fatty Acid Composition of Suckling Lambs of the Protected Geographical Origin Abbacchio Romano"

_animals, 2019, doi:10.3390/ani10010025_

Round 1

Reviewer 1 Report

Animals  Fusaro et al., 2019

General comments

The paper is within the scope of the journal and represents useful research on the nutritional effects of maternal diet on the quality of suckling lambs.

The main limitations of the paper are reported below.

The paper seems to ignore several other papers previously published in international journals on which linseed or linseed oil was fed to ewes and the effects of the suckling lambs were reported. The list of some of them is reported at the end of the comments. Of course, the objectives and the discussion should be revisited in light of the published literature.

The methodology used to sample the milk in mothers that were suckling their lambs is not reported. Similarly, no information is given on the methods used to assess the milk production of the mothers.

The English is in some parts poor and misleading.

Specific comments

L2 and 3: Abbacchio Romano is not a breed, thus its meaning should be clear in the title. i.e. …suckling lambs of the protected geographical origin Abbacchio Romano.
Besides this, I do not see in the experiment anything really specific to this PGI and I think the results can be extended to any similar feeding situation, considering the no breed is specified in the paper. Thus, the PGI can be taken out from the title

L34: see comments for L2 and 3.

L 48:  ….rigorous certified production ….

L 49: cut THIS and put ITS

L 54: add DAIRY before SHEEP

L55:  cut BRED and put FED

L56:  8 kg? 28-40 d old suckling lambs can easily reach 10-12 kg, as also shown by your experiment. Same line: too many AND. Change as follows (e.g.): days of age, generating…. Add DAIRY SHEEP before farmers

L 58: Academy of Nutrition and Dietetics, ..

L67 …oils, in animal diets…

L 73: farm ration? It can be anything. Too generic. Indoor fed dry ration?

L83:

L 83: ADMINISTRATION is probably more appropriate for drugs. I would use SUPPLY. Same row:  PARTURITION, not birth

L90: what does mean farm group? All the ewes were kept in the same farm. Change the name of the treatment (e.g.): Control, unsupplemented TMR…

L92: here you use grams, in the other parts kg. Be consistent throughout the paper

L 93: concentrates are supplements! clarify

L96: g and kg in the same line! Be consistent. Is the linseed included in the 800 g/ of concentrate? clarify

L103: how did you avoid that the lambs would eat mother’s feed?

TABLE 1: how did you sample the pasture to assess its composition? Method? Frequency?

L109: how did you collect the milk samples: complete mammary emptying? At T =0 there is not milk but colostrum. Did you sample it?

L 169 No information on group fed forage intake in the L and F group?

L 171? How did you measure milk production?? When?? How did you feed the lambs in the meanwhile? Clarify the methods

L172: milk samples: do you mean treatment means? End values? Please clarify

L 173-175. The whole sentence is very confused. Please clarify.

Table 2: Units? The title should include also the word YIELD. What is the meaning of the numbers for yield? 9.4 of what? kg per week? Per group?

Figure 1a : Day-dependENT…why the figures are all number one and then letters (a1, 1b, etc.). The guidelines suggest to number them in sequence (1,2,3..)

                   Why do you have in the figure both asterisks and letters to indicate significant differences?

L188 day-dependENT (here and in all captions of the figures)

Figure 1b, 1c = dependENT why not Figure 2 and 3? The symbols and lines are different compared to figure 1c. Please be consistent in ALL figures when you assign symbols and lines to the treatments. Why do you have in the figures both asterisks and letters to indicate significant differences? Similar problems also for the other figures. Adjust them ALL

L213 = signifiCant

Table 3: SEM 0.45 (dot missing)

Table 4. Notes: P  should be capital letters

L243: different THAN

L245: than THE animals…

L281: please consider in the discussion the results of the other publications in which linseed was used in the mothers or in lactating ewes. They can give important contributions to explain your results.  Some are listed below, make sure there are no other in the literature. This is your duty, I am just suggesting some I easily found.

L283: …did not differ (SIGNIFICANTLY is optional ) among the ewes fed with the three ….

Fed with is incorrect, here and in the rest of the paper: fed the diets…

L285: designed  for ?? used by

L289 …linseed COMPARED TO….

L292: owing to THE lipids USED IN THE RATIONS.

L295: we DID not observe any effect… or…there were no effects…

L296 =with

L299 = findings in sheep… citations????

L299 =CONTENT is present twice  ..compared with grazing ewes : compared with that of the pasture grazed by the ewes

L312 : make sure you integrate the discussion on FA with all relevant the papers available

L308-311 = I would suggest other motivations: a) linseed increased milk fat and decreased milk protein as often happens with protein sources b) pasture-fed ewes had low fat and very high protein probably because the NDF of the pasture stimulated much less than the hay the rumination activity and the saliva production (lower milk fat) and because the sugar content of the pasture might have stimulated microbial activity and thus high milk protein. In the methods, you mentioned milk urea but you did not report it. It might give clear indications of what happened.

L346.: increasED

L 347: the effect of linseed meal in sheep milk was reported in other publications as well.

L 358 : unclear. I would suggest: dairy ewes were grazing pasture in the vegetative stage and rich in PUFA and this probably enhanced the PUFA in their milk.

Figure 1d: see comments for the other figures

L372: yield production of what? Milk or meat? Lambs do not produce milk.. clarify the subject and the product. Same line:  Even THOUGH , not even when.

L369-375: I think that the motivation is different. Since all the ewes had only singles (how many males and how many females for each treatment?), the milk produced by the mothers could have been different among treatments but it was sufficient to maximize the growth of all lambs even in the treatment with lowest milk production. If milk is not all suckled, the mother adjusts quickly their production to the amount used by their lambs, so the amount you measured (how?? See methods) could have been affected by the low demand of milk.

L379 with

L399-400 I suggest: We considered that In the young lambs….

L406-407: do you mean the other feeds you used are unhealthy? Please be careful in this type of statements.

L410 : of which treatment you are talking about?   Same line: ….the results OF a previous..

L413  MORE, not better

L415 by THE scientific…

L426 and 427: it should be considered  ......we considered… chose between personal and impersonal and be consistent. Too many “considerations”, simplify

L431 : compared to      cut  more so than

L329 administering ?? you did not supply it, it was sucked by the lambs! Just cut it. The milk NATURALLY enriched with CLA…. Why 8kg= your lambs were heavier! (more than 10 kg)

L458 please change. See comments for the title.   …the FA composition of the PGO origin Abbacchio Romano suckling lambs…

L463: LOWER,  not higher

SUGGESTED ADDITIONAL LITERATURE

Berthelot V.,  P.Basb, E.PottiercJ.Normandd.2012. The effect of maternal linseed supplementation and/or lamb linseed supplementation on muscle and subcutaneous adipose tissue fatty acid composition of indoor lambs. Meat Science Volume 90, Issue 3,  Pages 548-557

Borys B, A Borys, JJ Pajak. The fatty acid profile of meat of suckling lambs from ewes fed rapeseed and linseed. - Journal of Animal and Feed Sciences, 2005

Correddu, F., Gaspa, G., Pulina, G., Nudda, A. Grape seed and linseed, alone and in combination, enhance unsaturated fatty acids in the milk of Sarda dairy sheep
(2016) Journal of Dairy Science, 99 (3), pp. 1725-1735.

Gallardo, B., Manca, M.G., Mantecón, A.R., Nudda, A., Manso, T. Effects of linseed oil and natural or synthetic vitamin E supplementation in lactating ewes' diets on meat fatty acid profile and lipid oxidation from their milk fed lambs (2015) Meat Science, 102, pp. 79-89
Nudda, A., Atzori, A.S., Boe, R., Francesconi, A.H.D., Battacone, G., Pulina, G. Seasonal variation in the fatty acid profile in meat of Sarda suckling lambs. (2019) Italian Journal of Animal Science, 18 (1), pp. 488-497. 
Nudda, A., Battacone, G., Bee, G., Boe, R., Castanares, N., Lovicu, M., Pulina, G. Effect of linseed supplementation of the gestation and lactation diets of dairy ewes on the growth performance and the intramuscular fatty acid composition of their lambs (2015) Animal, 9 (5), pp. 800-809.
Nudda, A., Battacone, G., Neto, O.B., Cannas, A., Francesconi, A.H.D., Atzori, A.S., Pulina, G. Feeding strategies to design the fatty acid profile of sheep milk and cheese
(2014) Revista Brasileira de Zootecnia, 43 (8), pp. 445-456. 

Author Response

Response to Reviewer 1 Comments

General comments:

The paper seems to ignore several other papers previously published in international journals on which linseed or linseed oil was fed to ewes and the effects of the suckling lambs were reported. The list of some of them is reported at the end of the comments. Of course, the objectives and the discussion should be revisited in light of the published literature.

The methodology used to sample the milk in mothers that were suckling their lambs is not reported. Similarly, no information is given on the methods used to assess the milk production of the mothers.

The English is in some parts poor and misleading.

As the reviewer suggested we improved discussion and take into account the papers previously published in international journals on which linseed or linseed oil was fed to ewes and the effects of the suckling lambs were reported. The methodology used to sample the milk was clarified. In attachment the certificate of english editing by EDITAGE www.Editage.com

Specific comments

L2 and 3: Abbacchio Romano is not a breed, thus its meaning should be clear in the title. i.e. …suckling lambs of the protected geographical origin Abbacchio Romano.
Besides this, I do not see in the experiment anything really specific to this PGI and I think the results can be extended to any similar feeding situation, considering the no breed is specified in the paper. Thus, the PGI can be taken out from the title

Response:

L2 and 3: As the Reviewer suggested the title has been modified as follow:  “Effect of ewe diet on milk and muscle fatty acid composition of suckling lambs of the protected geographical origin Abbacchio Romano”

L34: see comments for L2 and 3.

L34: “Abbacchio Romano” has been deleted

L 48:  ….rigorous certified production ….

L48: Changed as requested.

L 49: cut THIS and put ITS

L49: Changed as requested.

L 54: add DAIRY before SHEEP

L54: Changed as requested.

L55:  cut BRED and put FED

L55: Changed as requested.

L56:  8 kg? 28-40 d old suckling lambs can easily reach 10-12 kg, as also shown by your experiment. Same line: too many AND. Change as follows (e.g.): days of age, generating…. Add DAIRY SHEEP before farmers

L56: Changed as requested.

L58: Academy of Nutrition and Dietetics, ..

L58: Changed as requested.

L67 …oils, in animal diets…

L67: Changed as requested.

L 73: farm ration? It can be anything. Too generic. Indoor fed dry ration?

L 73: Changed as requested.

L 83: ADMINISTRATION is probably more appropriate for drugs. I would use SUPPLY. Same row:  PARTURITION, not birth

L83: Changed as requested.

L90: what does mean farm group? All the ewes were kept in the same farm. Change the name of the treatment (e.g.): Control, unsupplemented TMR…

L90: Changed as requested.

L92: here you use grams, in the other parts kg. Be consistent throughout the paper

L92: Changed as requested.

L93: concentrates are supplements! Clarify

L93: The sentence has been modified as follow: The ingredients of the total mix ration (TMR) were grass hay at 1,100 and 0,800 kg/day of concentrate-based meal (oat, barley, and soybean).

L96: g and kg in the same line! Be consistent. Is the linseed included in the 800 g/ of concentrate? Clarify

L 96:Changed as requested.

L103: how did you avoid that the lambs would eat mother’s feed?

L103: Lambs were maintained permanent with their dams and had access to their dam’s forages, but did not have access to the concentrates. The term “exclusively” in the sentence was removed.

TABLE 1: how did you sample the pasture to assess its composition? Method? Frequency?

TABLE 1: Hand plucked pasture samples were taken three times a week in the afternoon as described by Cabiddu et al., 2017.

L109: how did you collect the milk samples: complete mammary emptying? At T =0 there is not milk but colostrum. Did you sample it?

L 109: The sentence was modified as follow:

Individual milk samples were collected from each ewe by parlour-milking at the beginning of the experimental period starting on day 4 (T0), 7 and 14 days after lambing until day 28. On each time sample point dams were separated from their lambs two hours before milking as the morning and in afternoon. The milk collected was sampled and refrigerated until the analysis.

L 169 No information on group forage intake in the L and F group?

L 169 We have recorded the group intake for F and L but not for P. For this reason we did not show any information about feed intake.

L 171? How did you measure milk production?? When?? How did you feed the lambs in the meanwhile? Clarify the methods

L171: Milk yield was recorded starting from day 28 when dams were separated by their lambs. Milk yield was removed from Table 2.

L172: milk samples: do you mean treatment means? End values? Please clarify

172: Milk composition refers to samples obtained at the end of the experimental period.

L 173-175. The whole sentence is very confused. Please clarify.

L 173-175: Dietary treatments determined significant differences (P < 0.05) among milk total fat percentage from each group showing higher value for group L than F, and lower fat percentage in milk from group P.

Table 2: Units? The title should include also the word YIELD. What is the meaning of the numbers for yield? 9.4 of what? kg per week? Per group?

Table 2: Title and units modified as suggested. By mistake, Table 2 shows the weekly average production data recorded for each individual animal/group. The data was corrected showing the weekly production of each group.

Figure 1a : Day-dependENT…why the figures are all number one and then letters (a1, 1b, etc.). The guidelines suggest to number them in sequence (1,2,3..)

Figure 1a : Modified as suggested

Why do you have in the figure both asterisks and letters to indicate significant differences?

Significant differences in the Figures are now presented with letters.

 L188 day-dependENT (here and in all captions of the figures)

L188 Modified as suggested

Figure 1b, 1c = dependENT why not Figure 2 and 3? The symbols and lines are different compared to figure 1c. Please be consistent in ALL figures when you assign symbols and lines to the treatments. Why do you have in the figures both asterisks and letters to indicate significant differences? Similar problems also for the other figures. Adjust them ALL

Modified as suggested

L213 = significant

L213: Modified as suggested

Table 3: SEM 0.45 (dot missing)

Table 3:Modified as suggested

Table 4. Notes: P  should be capital letters

Modified as suggested

L243: different THAN

L243: Modified as suggested

L245: than THE animals…

L245: Modified as suggested

L281: please consider in the discussion the results of the other publications in which linseed was used in the mothers or in lactating ewes. They can give important contributions to explain your results.  Some are listed below, make sure there are no other in the literature. This is your duty, I am just suggesting some I easily found.

L 281: The additional literature follows:

Fattore, E., Massa, E. Dietary fats and cardiovascular health: A summary of the scientific evidence and current debate. International J. of Food Sciences and Nutrition 2018, 69, 916-927.

Nudda A, Battacone G, Bee G, Boe R, Castanares N, Lovicu M, Pulina G. Effect of linseed supplementation of the gestation and lactation diets of dairy ewes on the growth performance and the intramuscular fatty acid composition of their lambs. Animal. 2015, 9, 800–809.

Nudda A, Battacone G, Boe R, Manca MG, Rassu SPG, Pulina G. Influence of outdoor and indoor rearing system of suckling lambs on fatty acid profile and lipid oxidation of raw and cooked meat. Ital J Anim Sci. 2013, 12, 459–467.

Nudda A, Battacone G, Boaventura Neto O, Cannas A, Francesconi AHD, Atzori AS, Pulina G.. Feeding strategies to design the fatty acid profile of sheep milk and cheese. R Bras Zootec. 2014, 43, 445–456.

Gallardo B, Manca MG, Mantecon AR, Nudda A, Manso T. Effects of linseed oil and natural or synthetic vitamin E supplementation in lactating ewes’ diets on meat fatty acid profile and lipid oxidation from their milk fed lambs. Meat Sci. 2015, 102, 79–89

Borys, A., Borys, J., PajÄ…k, J. The fatty acid profile of meat of suckling lambs from ewes fed rapeseed and linseed. J. of Animal and Feed Sciences, 2005. 14, 223 –226.

Berthelot, V., Bas, P., Pottier, E., Normand, J. The effect of maternal linseed supplementation and/or lamb linseed supplementation on muscle and subcutaneous adipose tissue fatty acid composition of indoor lambs. Meat Science 2012, 90 (2) 548–557

Priolo A, Didier, M., Jacques, A. Effects of grass feeding systems on ruminant meat colour and flavour. A review. Anim. Res. 2001 50 185–200

Nudda, A., Palmquist, D.L., Battacone, G., Fancellu, S., Rassu, S.P.G., Pulina, G. Relationships between the contents of vaccenic acid, CLA and n−3 fatty acids of goat milk and the muscle of their suckling kids. Livestock Science, 2008, 118 195–203

Cantani, A. A home-made meat-based formula for feeding atopic babies: a study in 51 children. 2006. Eur Rev Med Pharmacol Sci. 2006,10:61–68.

Cardi, E., Corrado, G., Cavaliere, M., Frandina, G., Pacchiarotti, C., Rea,P., Mazza, M.L., Nardelli, F., Agazie, E. Rezza-Cardi’s diet as dietary treatment of short bowel syndrome.Gastroenterology.,1998, 114:A-869.

Martino,F., Bruno, G., Aprigliano, D., Agolini, D., Guido, F., Giardini,O., Businco, L.. Effectiveness of a home-made meat based formula (the Rezza-Cardi diet) as a diagnostic tool in children with food-induced atopic dermatitis. 1998 Pediatr Allergy Immunol. 9:192–196.

L283: …did not differ (SIGNIFICANTLY is optional ) among the ewes fed with the three ….

L283: Modified as suggested

Fed with is incorrect, here and in the rest of the paper: fed the diets…

Modified as suggested

L285: designed  for ?? used by

L285: Modified as suggested

L289 …linseed COMPARED TO….

L289: Modified as suggested

L292: owing to THE lipids USED IN THE RATIONS.

L292: Modified as suggested

L295: we DID not observe any effect… or…there were no effects…

L295: Modified as suggested

L296 =with

L296:Modified as suggested

L299 = findings in sheep… citations????

L299: Modified as suggested

L302-303 =CONTENT is present twice  ..compared with grazing ewes : compared with that of the pasture grazed by the ewes

L302-303: Modified as suggested

L312 : make sure you integrate the discussion on FA with all relevant the papers available

L312 Modified as suggested

L308-311 = I would suggest other motivations: a) linseed increased milk fat and decreased milk protein as often happens with protein sources b) pasture-fed ewes had low fat and very high protein probably because the NDF of the pasture stimulated much less than the hay the rumination activity and the saliva production (lower milk fat) and because the sugar content of the pasture might have stimulated microbial activity and thus high milk protein.

L308-311: The sentence has been modified has follow Supplementation with extruded linseed increased milk fat and decreased milk protein as often happens with protein sources. The lower concentration of lignin and the sugar content of the pasture might have stimulated microbial activity and thus resulted in higher milk protein compared to to F and L.

In the methods, you mentioned milk urea but you did not report it. It might give clear indications of what happened.

In methods section milk urea is a misprint, so we deleted it in the paragraph.

L346.: increased

 L346: modified as suggested

L 347: the effect of linseed meal in sheep milk was reported in other publications as well.

L347: modified as suggested

L 358 : unclear. I would suggest: dairy ewes were grazing pasture in the vegetative stage and rich in PUFA and this probably enhanced the PUFA in their milk.

L 358: Modified as suggested

Figure 1d: see comments for the other figures

Figure 1d Modified as suggested

L372: yield production of what? Milk or meat? Lambs do not produce milk.. clarify the subject and the product. Same line:  Even THOUGH , not even when.

L372: Modified as suggested

L369-375: I think that the motivation is different. Since all the ewes had only singles (how many males and how many females for each treatment?), the milk produced by the mothers could have been different among treatments but it was sufficient to maximize the growth of all lambs even in the treatment with lowest milk production. If milk is not all suckled, the mother adjusts quickly their production to the amount used by their lambs, so the amount you measured (how?? See methods) could have been affected by the low demand of milk.

L369-375: modified as suggested The lambs in the present work were fed maternal milk and exhibited no differences in their performances (Table 3) even though the concentration of protein and fat in the milk of dams were different between the groups (Table 2).Previous research has shown that changes in suckling lamb performance are mainly related to differences in milk yield, milk fat, and protein levels [27]. in their growth performance . In our work, the time of suckling (28 days) was probably too short to have an effect on lamb performances owing to the different concentrations of fat and protein in the milk. Moreover, since all the ewes had only singles (the number of male and female lambs were balanced between treatments)  the milk produced by the mother could have been different among treatments with lowest milk production. If milk is not all suckled, the mothers adjust quickly their production to the amount used by their lambs, so the amount measured could have been affected by the low demand of milk.

L379 with

L379: modified as suggested

L399-400 I suggest: We considered that In the young lambs….

L399-400: modified as suggested

L406-407: do you mean the other feeds you used are unhealthy? Please be careful in this type of statements.

L406-407: The sentence has been modified explaining that it could be possible modifying fatty acids composition of meats from ruminants

L406-407:Sentence modified as follow: However, it is possible to reduce the concentration of SFA in ruminant meat products by enhancing polyunsaturated fatty acids concentration of PUFA in their rations.

L410: of which treatment you are talking about?   Same line: ….the results OF a previous..

L410: Modified as suggested

L413:  MORE, not better

L413: Modified as suggested

L415 by THE scientific…

L415: Modified as suggested

L426 and 427: it should be considered  ......we considered… chose between personal and impersonal and be consistent. Too many “considerations”, simplify

L426-427: Modified as suggested

L431: compared to      cut  more so than

L431 Modified as suggested

L439 administering ?? you did not supply it, it was sucked by the lambs! Just cut it. The milk NATURALLY enriched with CLA…. Why 8kg= your lambs were heavier! (more than 10 kg)

L439 Modified as follow: A period of 28 days was enough to guarantee the transfer of CLA from milk to the lamb meat.

L458 please change. See comments for the title.   …the FA composition of the PGO origin Abbacchio Romano suckling lambs…

L458 Modified as suggested

L463: LOWER,  not higher

 L463 Modified as suggested

Reviewer 2 Report

n my opinion, the article is correctly written, but the problem is its relevance, considering they worked only for 28 d and without considering physiological changes that happen in milk from ewes, regarding protein and fat content, from the birth until the moment of weaning. On the  other hand, I can understand that study was very local, focused on a little local market for 28 d-old lamb meat, however, the most of the countries sacrificed lambs when their meat content  aport an optimum % of bone, maximum of muscle and minimum of fat (more or less 4 mo old, showing variations according to the breed), being affected only by the diet quality after weaning, which was not considered in this study.

Author Response

Response to Reviewer 2 Comments

Comments and Suggestions for Authors

In my opinion, the article is correctly written, but the problem is its relevance, considering they worked only for 28 d and without considering physiological changes that happen in milk from ewes, regarding protein and fat content, from the birth until the moment of weaning. On the  other hand, I can understand that study was very local, focused on a little local market for 28 d-old lamb meat, however, the most of the countries sacrificed lambs when their meat content  aport an optimum % of bone, maximum of muscle and minimum of fat (more or less 4 mo old, showing variations according to the breed), being affected only by the diet quality after weaning, which was not considered in this study.

Response:

Consumers increasingly pay more attention to the lipid profile of meat products from ruminants. The main goal of the present trial was to obtain meat from ruminants naturally enriched with PUFA. In particular, we carried out survey aimed to evaluate muscle fatty acid composition of suckling lambs of the protected geographical origin Abbacchio Romano at 28 days of age after birth.

In this trial the meat FA profile was similar to the FA profile of the milk of their dams and this relationship has previously been described in suckling lambs [Scerra et al., 2007; Manso et al., 2011; Nudda et al., 2015;]. In lambs at 28 days of age, the rumen is not yet functional, so the milk FA are absorbed directly by the intestine without ruminal biohydrogenation activity. Supplementation with extruded linseeds in the diet of dams was able to increase the levels of PUFA, particularly n-3 FA, in lamb meat and could be a beneficial feeding strategy to apply in ewe farms when fresh pastures are not available.

Reviewer 3 Report

Consumer interest in food quality increases. For this reason, authors carried out survey aimed at meat quality of suckling Abbacchio Romano lambs. This study brings a very detail analysis. The results are innovative, reflecting actual topic and original, and therefore potentially reasonable for publication in this journal. However, there are some improvements that should be taken into consideration before making decision about accepting the manuscript.

General revision:
The introduction is strictly oriented into meat quality of suckling Abbacchio Romano lambs. This should be the main goal of this study. Information about milk analysis (milk production, milk solids or milk FAs distribution) are in these terms important and informative as it represents a unique source of nutrition for lambs. However, deep analysis of milk related to feeding regime independently is out of the defined aim, according to my opinion. These results should be presented in this study in form of base statistics and further related in context of lamb’s growth and lamb’s meat quality attributes. This could be done as e.g. correlation analysis between specific milk production and other milk traits and lambs meat characteristics additionally performed to the current evaluation of effect of mother’s diet. This extended evaluation will be much address, reflecting defined aims in introduction or conclusion, respectively. For this purpose, milk characteristics should be recounted as the milk, milk solids or FAs provided for the lambs during the observed period.

Specific revision:
L 109 specify milking procedure (hand vs. parlour milking, using e.g. oxytocin application etc.)
L 110 – 111 add units of milk solids and fatty acids (%, g …).
L 112 how is somatic cell count related to your results?
L 164 – 167 Statistical analysis of data should be more explained e.g. what factors were evaluated in model for milk analyses and why repeated measurements…
L 183 add units to the Table 2
L 187 use either letters or stars for significant differences (for all Figures)

Author Response

Response to Reviewer 3 Comments

Comments and Suggestions for Authors

Consumer interest in food quality increases. For this reason, authors carried out survey aimed at meat quality of suckling Abbacchio Romano lambs. This study brings a very detail analysis. The results are innovative, reflecting actual topic and original, and therefore potentially reasonable for publication in this journal. However, there are some improvements that should be taken into consideration before making decision about accepting the manuscript.

General revision:
The introduction is strictly oriented into meat quality of suckling Abbacchio Romano lambs. This should be the main goal of this study. Information about milk analysis (milk production, milk solids or milk FAs distribution) are in these terms important and informative as it represents a unique source of nutrition for lambs. However, deep analysis of milk related to feeding regime independently is out of the defined aim, according to my opinion.

These results should be presented in this study in form of base statistics and further related in context of lamb’s growth and lamb’s meat quality attributes. This could be done as e.g. correlation analysis between specific milk production and other milk traits and lambs meat characteristics additionally performed to the current evaluation of effect of mother’s diet. This extended evaluation will be much address, reflecting defined aims in introduction or conclusion, respectively. For this purpose, milk characteristics should be recounted as the milk, milk solids or FAs provided for the lambs during the observed period.

As the reviewer highlighted,  the main goal of this study was meat
quality of suckling Abbacchio Romano lambs. For this reason we
deleted much of the discussion about the milk production and the milk
related to feeding regime. Moreover, we improved discussion on the milk quality composition in context of lamb’s growth and lamb’s meat quality attributes. However, lambs meat quality has been discussed in this experimental research starting from milk fatty acid composition during the 28
days of suckling as reported by other authors (e.g. Scerra et al., 2007; Gomez Cortes et al., 2014; Lobon et al., 2018)

Specific revision:
L 109 specify milking procedure (hand vs. parlour milking, using e.g. oxytocin application etc.)

L 109 Individual milk samples were collected from each ewe by parlour-milking at the beginning of the experimental period starting on day 4 (T0), 7 and 14 days after lambing until day 28.

L 110 – 111 add units of milk solids and fatty acids (%, g …).

L 110 – 111 Modified as suggested
L 112 how is somatic cell count related to your results?
L 112 Somatic cell count has been a typing mistake in the "Material Methods" section.

L 164 – 167 Statistical analysis of data should be more explained e.g. what factors were evaluated in model for milk analyses and why repeated measurements…
L 164 – 167 As suggested we have better explained Statistical analysis.

Data on FA composition of milk were analyzed by mixed model repeated-measures GLM of the SPSS version 13.0 statistical package (SPSS, 2006), including in the model the fixed effects of dietary treatments, sampling time, and the interactions between them. The data collected on carcasses and meat were processed by GLM in SPSS considering the fixed effect of dietary treatment. Tukey’s test was used to assess significant differences between treatment means.

Milk chemical composition was analyzed by GLM procedure considering dietary treatment as fixed effect. Data on FA composition was analyzed by repeated measurements because we consider as fixed effects dietary treatments and sampling time during the experimental period.

L 183 add units to the Table 2

L 183 Modified as suggested
L 187 use either letters or stars for significant differences (for all Figures)

L 187 Modified as suggested

Reviewer 4 Report

Observations and recommendations were included in the attached file

Author Response

Response to Reviewer 4 Comments

The research is of scientific interest and relevant from the point of view of consumer health.

Materials and methods

L135-137: Describe more details of the color measurement: aperture of observer, type of illuminant used.

Meat colour was measured using a Minolta Chromameter (Minolta CR-300, Tokyo, Japan) with a D65 illuminant and an 8-mm aperture.

The following text (L155-163) should be excluded from the statistical analysis section and inserted into the fatty acid quantification section:

..the data on FA composition were processed to calculate the following FA classes and indices: MUFA (FA with single double bond), PUFA (FA with more than one double bond); SFA (FA without double bonds); UFA (FA with one or more double bonds); n-3 (SC18:2 t11, c15 + C18:2 c9, c15 + C18:3 c9, c12,c15 + C:22:5 c7, c10, c13, c16, and c19 + eicosapentaenoic acid (EPA) + docosahexaenoic acid (DHA)); n-6 (S C18:2t9, t12 + C18:2 c9, t12 + C18:2 t9,c12 + C18:2 c9, c12 + conjugated linoleic acids (CLA) t10, c12 + C20:2 c11, c14 + C20:3 c8, c11, c14 + C20:4 c5, c8, c11, and 160 c14). The atherogenic index was calculated according to Ulbricht and Southgate [13] as follows: (C12:0 + 4 × C14:0 + C16:0)/(MUFA + PUFA). The I-Harris index was calculated as the sum of EPA and DHA [14], and undesired trans FA (UTFA) as the sum of C18:1t9 + C18:2 t9t12 [15].

L155-163: Modified as suggested

L165: For the means comparison, what test was used? Tukey-Kramer? Duncan? Fisher? Indicate.

L165: Tukey’s test was used to determine significant differences between the means.

Results

L173-175: Rewrite.

L173-175:The sentence has been modified

Figures 1a, 1b, 1c, y 1d, should be identified as Figure 1, 2, 3, and 4, respectively.

Figures: Modified as suggested

Within the figures do not use "*" to indicate the P value, just the uppercase letters.

Modified as suggested

L220-225: The paragraph: Meat and fat characteristics in response to the three dietary treatments are shown in Table 4. Meat and fat lightness L* were similar in the three groups, while a* was comparatively higher in group P than in groups F and L (P < 0.001) for meat and fat (P < 0.05). Another factor which was not influenced by dietary treatment was pH at both 45 min and 48 hours after slaughtering (P > 0.05). The drip and cooking losses were also not significantly different among the experimental groups., should be excluded from this section (3.2) and create a new section (3.4) "meat quality of suckling lambs".

Modified as suggested

In results tables, include the P-values to three digits, whether or not the effect of treatments significative.

Modified as suggested

Table 5: suggested title: "Sum of fatty acids (expressed in g/100 g) and nutritional values of meat....".

Modified as suggested

Round 2

Reviewer 1 Report

The authors markedly improve the paper.
I suggest only few corrections:

L110-111: it is not clear for how many times (sampling days) the milk was collected  L 111-112: the sentence is unclear. I suggest to modify it as follows:
On each milk sampling day, the ewes were separated from their lambs two hours before milking both in the morning and in the afternoon.  L 169: space among words L 282: among, not between L 295: comma after the parenthesis, not before L 325: not dot after WELL L 369: comma after the parenthesis, space at the end of the sentence L 433: consider

Author Response

L110-111: it is not clear for how many times (sampling days) the milk was collected 

Individual milk samples were collected from each ewe by parlour-milking at the beginning of the experimental period starting on day 4 (T0), 7, 14 and 28 after lambing.

L111-112: the sentence is unclear. I suggest to modify it as follows:
On each milk sampling day, the ewes were separated from their lambs two hours before milking both in the morning and in the afternoon. 

L119-120 that has been modified as suggested.

L 169: space among words

L169: Modified as suggested

L 282: among, not between

L282: Modified as suggested

L 295: comma after the parenthesis, not before

L295: Modified as suggested

L 325: not dot after WELL

L325: Modified as suggested

L 369: comma after the parenthesis, space at the end of the sentence

L369: Modified as suggested

L 433: consider

L433: Modified as suggested

Reviewer 3 Report

I appreciate correction in the manuscript. All specific revisions have been explained and added. However, the inconsistency in milk evaluation – as I noticed before – still remains. The idea that differences in ewe diet influence lamb’s growth traits or lamb’s meat characteristics throughout received milk is correct and logical. So, if you want to make milk analysis for this purpose it can be done. However, it simply must be explicitly obvious and supported by literature in Introduction and it must be reflected in aims of the study. So, add adequate literature supporting reason for milk analysis in introduction, correct aims, and let the evaluation as it is. The other possibility is to reduce part concerning milk analysis on level of base statistics (just to illustrate mechanism of ewe diet on lamb’s growth throughout mothers’ milk). Or remove part concerning milk evaluation from your manuscript and publish it independently in detail in another manuscript. Results describing effect of ewes’ diet differences on their lamb’s meat growth traits and meat quality characteristics are informative and strong enough.

Author Response

As the Reviewer suggested we have improved literature in the introduction to better explain how the  differences in ewe diet influence lamb’s growth traits and  lamb’s meat characteristics. We corrected also the aims of the study.

In the following section the new version of introduction

In recent years, consumers have stimulated research for producing quality products guaranteed by rigorous certified production, certificates, and methods that respect the environment. To protect its gastronomic and cultural heritage in a global market, the European Union has established two protection systems known as protected designation of origin (PDO) and protected geographical indication (PGI). These systems were first regulated by the EC Regulation n. 2081/92 and then by Regulation EC no. 510/2006. In Italy, there are two lamb products registered under PGI: Sarda Suckling Lamb and Roman Lamb (Abbacchio Romano). The Roman Lamb represents 11–15% of the national meat market; they are slaughtered very young at about 10-12 kg after a suckling period of 20–30 days, in order to minimize the milk loss for cheese production and also because Italian consumers prefer this kind of product. From birth to slaughter, lambs are managed with their mothers and are fed almost exclusively on milk. Suckling lambs are functionally non-ruminants, and their meat FA profile should reflect the FA profile of the suckled milk [1,2]. Therefore, changes in milk FA composition due to supplements in the dam diet can induce important differences in the FA profile of the meat and fat depots of the suckling lamb [3,4]. Several strategies have been tested in recent years to improve the fatty acids (FA) profile of ewe fat, like the use of fresh pasture or linseed supplementation. Fresh pasture has been shown to be an excellent source of polyunsaturated fatty acids (PUFA) to increase these FA in milk [5] and subsequently in suckling lamb meat fat [6]. Unfortunately, in Italy like in others Mediterranean country, fresh pasture is not available during all the year and in this case linseed supplementation (oil or seed) is a reliable alternative feeding strategy to enrich with PUFA, Conjugated Linoleic Acid (CLA) or omega three (n-3) the milk fat from ewes [7,5,8] and for their suckling lambs.

Currently, consumers are aware that consuming lamb meat with high levels of saturated fatty acids (SFA) is negative from a nutritional perspective as advised by the Academy of Nutrition and Dietetics, which, from a dietetics perspective, promotes the reduction of SFA and favours higher polyunsaturated FA (PUFA) content [9]. This is based on clinical studies which have referenced the relationship between the presence of some dietary acids, such as medium-chain saturated fats (C12: 0, C14: 0, and C16: 0), serum cholesterol, and heart disease [10]. In general, the FA composition of lamb includes as much as 53% SFA, 34% monounsaturated FA (MUFA), and only 13% PUFA [10]. Livestock production shows a growing interest in improving the nutritional characteristics of the products, e.g. by natural enrichment of the animal diet, which leads to food characteristics that are beneficial to human health [11]. This natural enrichment could be achieved through pasture management, including “grass finished” animals, or the addition of supplements, such as flaxseed and oils in animal diets. In recent years, several studies have investigated the effect of ewe diet on the meat quality of their suckling lambs [12] showing how the intramuscular FA composition of suckling lambs is partly related to the sheep feeding system [13] during gestation or lactation [14,15,16] or both. To the best of our knowledge, there are no studies that have compared the suckling lamb meat obtained from grazing dams and that from animals fed indoors with a supplemented or un-supplemented diet.

The aim of this trial was to investigate whether the supplementation of dam diet with extruded linseed would be an alternative to pasture strategy for improving the intramuscular and subcutaneous FA compositions of their suckling lambs, without detrimentally affecting animal performance.
